# Targeting the DNA Damage Response to Overcome Cancer Drug Resistance in Glioblastoma

**DOI:** 10.3390/ijms21144910

**Published:** 2020-07-11

**Authors:** Alessandra Ferri, Venturina Stagni, Daniela Barilà

**Affiliations:** 1Department of Biology, University of Rome Tor Vergata, 00133 Rome, Italy; alessandraferri4@gmail.com; 2Laboratory of Signal Transduction, IRCCS-Fondazione Santa Lucia, 00179 Rome, Italy; venturina.stagni@cnr.it; 3Institute of Molecular Biology and Pathology, National Research Council (CNR), 00185 Rome, Italy

**Keywords:** DNA damage response, glioblastoma, DDR inhibitors

## Abstract

Glioblastoma multiforme (GBM) is a severe brain tumor whose ability to mutate and adapt to therapies is at the base for the extremely poor survival rate of patients. Despite multiple efforts to develop alternative forms of treatment, advances have been disappointing and GBM remains an arduous tumor to treat. One of the leading causes for its strong resistance is the innate upregulation of DNA repair mechanisms. Since standard therapy consists of a combinatory use of ionizing radiation and alkylating drugs, which both damage DNA, targeting the DNA damage response (DDR) is proving to be a beneficial strategy to sensitize tumor cells to treatment. In this review, we will discuss how recent progress in the availability of the DDR kinase inhibitors will be key for future therapy development. Further, we will examine the principal existing DDR inhibitors, with special focus on those currently in use for GBM clinical trials.

## 1. Introduction

The DNA damage response (DDR) is a collective term that gathers all the mechanisms that detect DNA damages, signal them and either promote their repair or trigger cell death pathways [1,2]. It has evolved as a protective mechanism to maintain our genetic information unchanged between generations, however, in a cancer therapy context, the DDR can be considered as a negative feature [3]. Indeed, under physiological conditions, the DDR protects our genome by removing errors and avoiding the insurgence of mutations. On the other hand, in tumors treated with DNA damaging agents, efficient DNA repair systems become the major cause for treatment failure [4].

Signalling pathways regulated by the DDR are numerous and partially overlap [3]. This orchestra is responsible for processing the two main types of DNA lesions: single-strand breaks (SSBs) and double-strand breaks (DSBs) [4]. At the center of DNA damage signalling, in response to DSBs, are the phosphoinositide 3-kinase-related kinases (PIKK) ATM, ATR and DNA-PK [5]. Activation of ATM/ATR/DNA-PK by DNA damage in turn results in phosphorylation of several substrates that control various pathways involved in DNA repair, checkpoint activation, apoptosis and transcription regulation. For example, ATM and ATR activate the checkpoint kinases Chk1 and Chk2 which then phosphorylate and inactivate Cdc25 and regulate cell cycle progression [5]. Instead, SSBs can result from endogenous oxidative damage, defective activity of cellular enzymes or erroneous incorporation of ribonucleotides in DNA [3]. Repair can occur through base excision repair (BER), involving poly (ADP-ribose) polymerase-1 and 2 (PARP1 and PARP2). PARP1 and PARP2 are crucial proteins for BER and act as sensors of SSBs promoting the recruitment and activation of critical downstream SSB repair effectors [3].

Accumulating evidence demonstrated that aberrant activation of DDR proteins (ATM, ATR, DNA-PK, Chk1, Chk2 and PARP) in cancer is strongly correlated with resistance to genotoxic anti-tumor therapeutics of cancer cells [3]. For this reason, DDR inhibitors are promising candidates in cancer treatment (Figure 1). More interestingly, DDR inhibitors have the potential to elicit synthetic lethal effects. Cancer cells with defects in one DDR pathway often depend on other pathways for their survival, and targeting these pathways of reliance can be exploited to cause selective cancer cell death [2].

In this review, we will briefly discuss the DNA damage response and how it can drive therapy resistance, with particular attention to glioblastoma (GBM). The DNA repair systems have already been extensively described elsewhere [1,5]. Here, we will mainly focus our attention on the three apical kinases of the DNA damage response, namely ataxia-telangiectasia-mutated (ATM), ataxia-telangiectasia- and Rad3-related (ATR) and DNA-dependent protein kinase (DNA-PK), and analyze how they can contribute to therapy resistance. Further, we will enumerate the major inhibitors that have been developed for each kinase, with special attention to those that have entered clinical trials that include GBM patients (see Table 1).

Finally, we will dedicate a paragraph to PARP inhibitors, which were the firsts DDR inhibitors (DDRi) to be developed and are currently the most studied either alone or in combination with other forms of treatment.

## 2. Glioblastoma Multiforme

Glioblastoma multiforme is the most aggressive of brain tumors. Despite early treatment, patients’ survival remains dismal, with an approximate life expectancy of 14 months [6]. Standard care after surgery consists of a combinatory therapy of ionizing radiations (IR) and temozolomide (TMZ). Both IR and TMZ act by damaging the DNA. IRs directly cause irreversible clustered DNA damage, generating interstrand crosslinks (ICL), single- or double-strand breaks leading to cell death [7]; on the other hand, TMZ has a more intricate mechanism of action. It can produce up to 12 different lesions, the most frequent being: N7-methylguanine (N7MeG) (80–85%), N3-methyladenine (N3MeA) or N3-methylguanine (N3MeG) (8–20%). Despite their frequency, these lesions are not by far as lethal as the minor O6-methylguanine (O6MeG) lesion (8%) [8]. The formation of this adduct causes the guanine to mispair with thymine, therefore activating the mismatch repair system (MMR). The MMR removes the incorrect thymine creating a gap. When the polymerase tries to seal the gap, it misreads the O6MeG once again and pairs it with a new thymine, thereby initiating a so called “futile repair cycle” [9,10]. After the second round of repair, replication forks stall and collapse, generating a double-strand break (DSB) and activating proteins of the DDR [11]. An alternative line of treatment is represented by chloroethylating nitrosoureas (CNUs), which act in a similar way producing 06-chloroethylguanine (O6-ClEtG) lesions [12].

Developing resistance to radiation and chemotherapy is a major clinical issue: a great part of resistance is caused by GBM’s enhanced activity of DNA repair systems that render DNA damaging treatments ineffective [13].

Developing resistance to radiation and chemotherapy is a major clinical issue: glioblastoma is a highly radio- and chemo-resistant tumor. Radio-resistance and chemo-resistance can be defined as the capacity of cancer cells to resist respectively to ionizing radiations or chemotherapeutic drugs. These characteristics render GBM a very difficult tumor to treat [3]. Although the molecular basis for its innate resistance has not been fully elucidated, a great part of it has been linked to the presence of cancer stem cells which are responsible for cancer initiation and recurrence and are characterized by the aberrant functionality of the DNA repair machinery. Indeed, GBM’s ability to resist DNA insults is directly attributable to its innate upregulation of DNA repair pathways which render treatments ineffective [13].

## 3. The DNA Damage Response in Mediating Resistance in Glioblastoma

Cells are constantly exposed to either exogenous or endogenous stresses that can lead to DNA damage and threaten the genomic integrity of the cell. As a form of protection, mammals have evolved several types of DNA repair systems, each specific for a precise type of damage [1].

Broadly, these repair systems can be divided in two groups: those repairing SSBs and those handling DSBs. Common cancer therapy includes radiation and chemotherapy, which trigger the formation of both single-strand and double-strand breaks, therefore stimulating the response of both types of repair mechanisms.

### 3.1. Single Strand Break Repair Systems

Single-strand break (SSB) repair systems comprehend base excision repair (BER), nucleotide excision repair (NER) and mismatch repair (MMR). All these systems operate by firstly recognizing and removing the damaged portion of the DNA. They then use the other strand as a template to synthetize a new correct sequence and seal the gap [1].

SSB repair systems are significantly activated after chemotherapy. As discussed above, administration of alkylating or chloroethylating drugs results in the formation of DNA adducts [8,12] that, upon detection by the repair machinery, can be repaired by either BER or NER accordingly to the type of damage. In a therapeutic context, the hope lies in the incapacity of the system to repair the induced lesions. Therefore, the activation of early repair systems such as BER and NER jeopardizes the attempt to turn the DNA adducts into greater forms of damage and lead cancerous cells to death [14].

It is otherwise important to specify that, while as a general rule, increased activation of DNA repair systems is counterproductive for standard cancer therapy, each type of treatment and repair mechanism must be considered individually. In fact, when discussing TMZ treatment, activation of BER and NER systems is not desirable. On the contrary, an operating MMR is instead required: with no active MMR, no “futile repair cycle” occurs and the DNA adducts are never converted into DSBs, which are ultimately what endangers the tumor [9,10].

The role of the MMR system in promoting TMZ sensitivity has also emerged in a recent CRISPR-Cas9 screening aimed at identifying genes that mediate TMZ resistance [15]. The screening was performed on patient-derived glioblastoma stem cells (GSCs) treated with high doses of TMZ. The results highlighted that three weeks after treatment, resistant cells displayed an enrichment of four genes involved in the MMR (MLH1, MSH2, MSH6 and PMS2) [15]. Of note, in recurrent GBM, a significant decrease in the MSH2, MSH6 and PMS2 protein expression level was reported [16], and reduction in MLH1 and PMS2 correlated with TMZ resistance [17].

### 3.2. Double Strand Break Repair Systems

When the damage is more extended and comprises both strands, the repair mechanisms are more complex. This is largely because when DNA information is missing from both strands, the polymerase has no template to copy to generate the new strand [1].

The two repair systems involved in double-strand break (DSB) repair are homologous recombination (HR) and non-homologous end-joining (NHEJ). The first one can only work if cells are damaged whilst in their late S/G2 phase of the cell cycle. This is because only during these phases the DNA has been replicated and each cell disposes of two copies of each DNA strand. Therefore, the HR enables the polymerase to use the sister chromatid as a template to synthetize the repaired strand [18,19].

When cells are damaged in different phases of the cell cycle and lack a second copy of the DNA, the NHEJ is called into action. The heterodimer KU 70/80 binds the DSB and recruits the DNA-dependent protein kinase catalytic subunit (DNA-PKcs). With no template to copy from, the NHEJ acts by joining the two ends of the DSB. This system is then able to repair the DSB but does not restore DNA to its original state before the damage [20].

The CRISPR-Cas9 screening cited above also identified genes involved with DSB repair mechanisms as essential for TMZ-resistance [15]. The spotted genes (e.g., MCM8 and MCM9) are connected to Fanconi anemia/interstrand crosslink and homologous recombination. In fact, knockout of MCM8 or MCM9 significantly increased TMZ sensitivity in GSCs, despite having no effect on non-tumor neuronal stem cells (NSCs) [15].

## 4. Temozolomide Treatment and Cell Death

The DNA damage response exerts its function first of all by regulating the cell cycle and at a second stage by triggering either repair or death signalling pathways. Among the first proteins to be recruited to the DSB, ATM and ATR kinases are key in sensing the damage and initiating a downstream signalling cascade [12,21].

As soon as DNA damage is detected, the cell cycle is arrested. One of the first goals of the ATR/ATM signalling cascade is in fact to activate checkpoint kinases 1 and 2 (Chk1 and Chk2) and slow down the cell cycle [1]. Once the cycle has been arrested, the cell attempts to repair the damage. However, if such damage is too extended, the prolonged activation of the DDR kinases spurs either apoptotic or necrotic death pathways [22].

### TMZ and CNUs Affect Cell Cycle and Death

TMZ creates lesions that cause replication forks to stall, activating in the first place the ATR-Chk1 axis. Only at a later stage is the ATM-Chk2 axis triggered, probably due to secondary effects caused by DSBs at collapsed replication forks [23,24]. It has been demonstrated that TMZ preferentially induces cells to arrest in the G2/M phase of the cell cycle: a recent work showed that 48h post-TMZ, GBM cells (i.e., T98G) are preferentially retained in this phase [25]. Similarly, another study had previously evidenced an increase in the percentage of G2/M cells upon TMZ treatment [26]. This G2/M arrest appears to be independent of p53 status as TMZ administration was shown to spur G2/M arrest both in p53-proficient (i.e., U87MG) and p53-deficient (i.e., E6-transfected U87MG and LN-Z308) GBM cell lines. The difference between p53-proficient and p53-deficient cells lays in the duration of this G2/M arrest, which is prolonged for p53-proficient cells [26].

Otherwise, it is also known that TMZ-induced ATM/ATR activation leads to p53 phosphorylation on serine 15 and 20, arresting the cell cycle in G1/S, thereby promoting transcription of pro-survival genes [27]. When damage is too extensive to be repaired, TMZ also promotes p53 phosphorylation on serine 46 which induces transcription of pro-apoptotic genes as FAS, BAK, BAX, PUMA and PTEN, preferentially leading to apoptotic cell death [28]. The G1/S arrested population is the one that undergoes apoptotic death, while the cells arrested in G2/M are more likely to become senescent if they have the wild-type p53 [25].

For both alkylating and chloroethylating drugs, p53 status is also key in deciding which death pathway to trigger. Gliomas expressing the wild-type p53 trigger the extrinsic pathway of apoptosis upon TMZ and the intrinsic pathway upon CNUs administration [29]. On the other hand, non-functioning p53 mainly results in the activation of the apoptotic intrinsic pathway upon TMZ and necrosis upon CNU administration [28].

## 5. Inhibition of DDR Kinases to Overcome Therapy Resistance in GBM

In the next paragraphs, we will consider the role of apical DDR kinases in repairing DNA damage and explore existing inhibitors that have been developed for these proteins. For each kinase, we will examine on-going or recently concluded clinical trials to underline the importance of DDR inhibitors in cancer therapy, especially for tumors that have a naturally hyperactive DDR, such as GBM.

### 5.1. ATM

ATM is a serine/threonine kinase belonging to the phosphatidylinositol-3 kinase-like kinase (PIKK) family. It is activated by several stimuli, including DNA DSBs, hypoxia and reactive oxygen species (ROS) and plays a dual role in cancer [30]. When both strands of the DNA are damaged, the sensor complex MRN (Mre11-Rad50-Nbs1) is engaged at the site of the damage. This in turn can recruit ATM, which becomes activated through an auto-phosphorylation process thereby initiating a signalling cascade. Among its main targets, there are Chk2 and p53. The activation of this signalling cascade leads to cell cycle arrest preferentially in the G1/S checkpoint [31], but also in the G2/M checkpoint [32] and consequently to either repair of the damage or initiation of death pathways.

ATM depletion results in an autosomal recessive disorder known as ataxia-telangiectasia which displays increased sensitivity to radiation. Knowledge of this suggested that inhibitors targeting ATM (ATMi) could be used as radio- or chemo-sensitizers, electing them among the firsts DDR inhibitors to be developed, starting from the small molecule ATP-analogue **KU55933 [33]**. This first-generation drug was a selective ATM inhibitor but lacked several fundamental drug attributes, such as good aqueous solubility and bioavailability and it was therefore soon substituted by its derivative **KU60019 [34]** and by **CP466722** [35]. Second-generation drugs, such as **AZ32,** brought considerable improvement to drugs’ availability and to their capacity to cross the brain–blood barrier (BBB) [36].

Pre-clinical studies have highlighted a promising effect for ATMi ameliorating patient’s response to radiations (or radiomimetic drugs) and topoisomerase I/II drugs (e.g., camptothecin, etoposide, doxorubicin) [33,34,35,36].

High-throughput screenings are also proving useful in the constant search for new molecules that could inhibit ATM. A cell-based screening mechanism, employing the in-cell Western (ICW) immunoassay, was used to identify two new molecules that could potentially act as ATM inhibitors: **SJ573017**, a potent Polo-like kinase 1 (PLK1) and 3 (PLK3), and **SJ573226**, a GSK-3 kinase inhibitor [37].

More recently, ATMi have entered Phase-I clinical trials either as monotherapy or in combination with other forms of treatment. As far as GBM is concerned, the ATM inhibitor **AZD1390** was specifically optimized to cross the BBB and has a good central nervous system (CNS) availability [38]. Its toxicity is currently being evaluated in a clinical trial combining AZD1390 and IR (see Table 1 CT-NCT03423628).

ATM plays a key role in DSB repair by homologous recombination (HR) as it triggers the formation of single-strand DNA overhangs that are key for HR initiation. Moreover, ATM loss is frequently reported in cancers and has been shown to cause a “mild” HR deficiency [39]. Synthetic lethality between ATM and ATR was revealed in several studies and spurred the usage of ATMi to sensitize HR-proficient cells to treatment [40].

As far as the role of p53 is concerned, contrasting studies have been published regarding the significance of p53 status for ATM inhibitor-induced toxicity. For example, studies conducted on human colorectal cancer cell lines showed p53 status to be inconsequential for ATMi KU59403 radio-sensitizing effect [41]. On the other hand, a study using GBM cell lines pointed out that KU60019 would render mutant p53 cells more sensitive to therapy than the correspondent p53-WT cell line [42]. Whether differences can be imputable on the different types of inhibitors used, the different ways in which sensitivity is determined or on the tumor type remains to be clarified.

The main target of ATM, Chk2, has also been under investigation for the generation of inhibitor molecules, but studies have so far only reached the pre-clinical stage, at least for GBM: **PV1019** [43] and **CCT241533** [44] were used in combination with IR, bleomycin or the PARP inhibitor olaparib in GBM cell lines. The least selective inhibitor **AZD7762** which targets both Chk2 and Chk1 reached a Phase-I clinical trial for GBM, but this had to be terminated early due to cardiotoxicity issues [45].

### 5.2. ATR

ATR is a serine/threonine kinase belonging to the PIKK family but, differently to ATM, it responds to single-strand DNA (ssDNA). Single-strand DNA is commonly generated when replication forks stall and collapse or during the processing of DSB sites [46]. Replication protein A (RPA) coats ssDNA to stabilize it and avoid the formation of destructive secondary structures. ATR works in couple with ATR-interacting protein (ATRIP), which is used to recruit ATR to RPA-coated ssDNA [46]. Once recruited to the site of the damage, regulatory complexes such as 9-1-1 (Rad9-HUS1-Rad1) stimulate ATR and drive its activation. Its main target is Chk1 which arrests the cell cycle in the intra-S or the G2/M phase [46].

The first ATR inhibitor (ATRi) to be developed, **VE-821**, was a competitive ATP-analogue that specifically inhibited ATR [47].

Both VE-821 [48] and its structural analogue AZD6738 [49] were used in pre-clinical trials on GBM cells and stem cells as monotherapy or in combination with cisplatin.

Second-generation drugs **VX-970** [50,51], **BAY1895344 [52]** and the AZ20 derivative **AZD6738** [49,53] acquired ameliorated selectivity and oral availability and were the first ATRi to enter clinical trials. There are currently several undergoing Phase-I/II clinical trials examining ATRi: AZD6738 is so far the only inhibitor suitable for glioma patients thanks to improved BBB-permeability [53].

As discussed above for ATMi, also ATRi are able to increase toxicity more powerfully in cancer cells lacking ATM or that display a mutated form of p53 [54,55,56]. Coherently, ATM deficiency is one of the most advantageous biomarkers used for patient’s selection in clinical trials employing ATRi [40,49,57].

It is otherwise interesting to note that ATRi have also been found to synergize with Chk1 inhibitors, increasing their toxicity, although the reason for this is still poorly understood [58].

Chk1 inhibitors as **SRA737** [59] and **LY2606368** [60] are being evaluated in Phase I/II clinical trials, not involving GBM patients so far. Their BBB-permeability is still unclear and under investigation, but a Phase-I clinical trial for **LY2606368** including medulloblastoma patients may lead the way for future studies in other brain tumors (see Table 1, CT-NCT04023669).

Synthetic lethality of ATRi or Chk1 has been proven also together with Wee1 inhibitors [61,62]. Wee1 is a kinase activated by Chk1 that inhibits cyclin-dependent kinase 1 (CDK1), thereby preventing mitotic entry [63].

Loss of the phosphatase CDC25A has been identified as a major cause for resistance to ATR inhibitors [64]. CDC25A loss causes cells to arrest before they undergo mitosis, thereby reducing the impact of DNA-damaging agents. A way around this form of resistance is represented by the concomitant use of Wee1 inhibitors. The only Wee1 inhibitor available so far is **MK1775** and is currently under Phase-I evaluation. It forces mitotic entry before replication has been completed resulting in abnormal mitoses and cell death [65,66]. It has a good BBB-permeability and is being tested in GBM patients treated with IR and TMZ [66,67,68].

### 5.3. DNA-PK

DNA-PK is once again a serine/threonine kinase belonging to the PIKK family. It is composed of the DNA binding heterodimer KU70/KU80 and the catalytic subunit called DNA-PKcs. It is a key protein in the non-homologous end-joining repair system, which as mentioned above, is required for the repair of DSBs that occur before DNA duplication [20].

Currently there are two DNA-PK inhibitors being tested in clinical trials: **M3814** [69,70] is being tested in a Phase-I clinical trial for advanced solid tumors and leukemia patients that have ATM deficiency. Indeed, DNA-PK inhibitors have previously been demonstrated to increase lethality in cancers with ATM loss [71,72]. **CC-115** is a recent compound which is able to cross the BBB and whose toxicity and efficacy are being evaluated in GBM patients, combined with IR and TMZ treatment [73].

### 5.4. PARP

Poly (ADP-ribose) polymerases (PARPs) are a family of 18 proteins united by their ability to transfer ADP-ribose to target proteins. PARP1 is the first and best studied protein of the family and shares 69% homology with PARP2. Both PARP1 and PARP2 are best known for their role in DNA repair mechanisms [74]. PARP proteins respond to SSBs and are involved in the base excision repair (BER) pathway, being part of the BER complex together with DNA ligase II, DNA polymerase β and XRCC1 [75]. Chemotherapeutic agents such as alkylating or chloroethylating agents typically induce a form of DNA damage that activates BER [3].

There is evidence for PARP upregulation in several types of cancers: hepatocellular carcinoma [76], breast cancer and ovarian cancer [77]. As far as GBM is concerned, PARP-1 was found to be more expressed in GBM biopsies than in healthy donor tissues [13,78]. Altogether, these findings suggest that inhibition of PARP activity may be beneficial for tumor treatment. In fact, PARP-deficient mice were found to be more sensitive to radiation and DNA-damaging agents [79].

Tumors that display impaired homologous recombination (HR) capacity, such as BRCA1-BRCA2-mutated tumors, were found to be strongly dependent on PARP-1 activity and therefore possibly more sensitive to its inhibition [80]. This is the reason why the first PARP inhibitors were specifically developed for ovarian or breast cancers displaying BRCA1/2 defects [81,82].

ARIEL2 and ARIEL3 were double-blinded Phase-II and Phase-III clinical trials evaluating the effect of the PARP inhibitor **Rucaparib** as monotherapy for ovarian cancer with HR deficiency [83]. Pre-clinical studies have focused on trying to extend Rucaparib use to GBM treatment in combination with TMZ. Despite having obtained promising results on GBM cultures, Rucaparib failed to improve TMZ sensitivity in orthotopic models, due to extremely poor CNS permeability [84]. On the other hand, **Niraparib** [85] and **Veliparib** [86] both showed good BBB-penetration and were approved for Phase-I/II clinical trials on GBM patients.

The GBM standard treatment drug temozolomide induces toxicity mainly through O6-MeG lesions because N3-MeA and N7-MeG lesions are easily repaired by BER [8]. Since PARP inhibitors are known to affect the BER system, they were thought to be potentially good TMZ sensitizers, by restoring the lethality of N3-MeA and N7-MeG adducts that make up 80% of TMZ-induced lesions. Whether TMZ sensitization by PARP inhibitors is due to their effect on the BER repair system or not is under investigation; knock-down of other BER proteins failed to further increase PARP-mediated TMZ sensitization, which was instead enhanced by knock-down of HR key proteins such as BRCA1 or Rad51 [87]. Another recent study pointed out a BER independent role for PARPi Veliparib and Olaparib in restoring TMZ sensitivity in cells that were resistant due to MMR deficiency [88].

At last, we discuss the PARP inhibitor **Olaparib** [89]. Olaparib is undergoing a Phase-I clinical trial for GBM in combination with radiation, TMZ and bevacizumab. Despite no evidence for Olaparib as a CNS available drug, it is being tested for brain tumors as well, as recent evidence has highlighted that many GBM patients show compromised integrity of the BBB and that in such cases, Olaparib may be successfully delivered to the tumor [90].

Many pre-clinical trials involving PARP inhibitors and TMZ have failed due to the high haematological toxicity induced by the combination of these drugs [91]. A solution could be presented by the Phase-I study PARADIGM-2 [92], which is studying dose-dependent toxicity in GBM patients that have been divided according to their MGMT expression. MGMT is a demethylase that removes TMZ-induced alkyl-adducts rendering TMZ treatment ineffective [93]. PARADIGM-2 patients that have methylated and poorly expressed MGMT are administered Olaparib plus radiation and TMZ, while TMZ-resistant patients with high levels of MGMT are treated with Olaparib plus a higher dose of radiation only [92].

## 6. Conclusions

Over recent years, the DDR has gained increasing importance in cancer treatment. Better understanding of the mechanisms regulating the DDR in cancer have helped clarify the role of the major DDR kinases in response to chemotherapy and radiation and have encouraged the development of DDRi as potential treatments. GBM is a multifaceted tumor that owes much of its incurability to the upregulation of DNA repair pathways that contrast therapy-induced DNA damage. All in all, when evaluating GBM treatment, many factors have to be kept in mind: HR efficacy, BER activity, MGMT status and drug BBB-penetrating ability being only part of the picture. What is certain is that the recent advances in DNA sequencing techniques and improvement of DDR inhibitors will enable a personalized form of treatment for patients, which may represent a valuable solution for a complex and variable tumor such as GBM.

**Table 1 ijms-21-04910-t001:** Summarizes the main inhibitors that target proteins of the DNA damage response. For each compound, existing clinical trials have been noted, focusing on those involving GBM patients where possible. Further, pharmacologically relevant characteristics such as the drug’s availability and blood–brain barrier (BBB) permeability have been highlighted.

Kinase	Inhibitor	Clinical Trial Phase	End Date	Drug Combinatory Strategy	Characteristics	BBB Permeability	Tumors/Cell Lines	Reference
**ATM**	KU55933	Pre-clinical	-	+IR +etoposide phosphate	Poor aqueous solubility and poor bioavailability	no	Human **cervical cancer** (HeLa), Human **osteosarcoma** (U2OS)	[33,94]
	KU60019	Pre-clinical	-	-	Poor aqueous solubility and poor bioavailability	no	Human **glioma** (U87, U1241)	[34]
	CP466722	Pre-clinical	-	+temozolomide	Improved aqueous solubility and bioavailability	no	Human **breast cancer** cell line (MCF7), **fibroblasts** (HFF), A-T cells, GBM12 **glioblastoma** xenograft cell lines	[35]
	KU59403	Pre-clinical	-	+irinotecan +etoposide phosphate	Improved aqueous solubility and bioavailability	no	Human **colon cancer** (HCT116, SW620), Human **osteosarcoma** (U2OS), Human **breast cancer** (MDA-MB-231)	[41]
	AZ32	Pre-clinical	-	+IR	Good bioavailability	yes	Human **glioma** (U87, LN18, T98G, …, A172)	[36]
	AZD0156	Phase-I NCT02588105 clinicaltrials.gov	30 April 2020	+olaparib +irinotecan	-	yes (poor)	Various metastatic **solid tumours** (including gastric adenocarcinoma, colorectal cancer)	[95]
	AZD1390	Phase-I NCT03423628 clinicaltrials.gov	05 April 2022	+IR	Good bioavailability	yes	Primary and recurrent **glioblastoma multiforme**	[38]
**ATR**	VE-821	Pre-clinical	-	+cisplatin	-	yes (poor)	Hamster **ovarian cells** (AA8), hamster **lung fibroblasts** (V79), human **glioblastoma multiforme** (M059J)	[47]
	NU-6027	Pre-clinical	-	+cisplatin +PARPi +hydroxyurea	-	unclear	**Breast** cancer, **pancreatic** cancer, **ovarian** cancer	[96]
	AZ20	Pre-clinical	-	monotherapy	Poor aqueous solubility	yes (poor)	**Colorectal adenocarcinoma** cell line (HT29), **glioblastoma** CSCs	[55,97]
	VX-970	Phase-I NCT02157792 clinicaltrials.gov	11 March 2020, 30 April 2025	monotherapy +cisplatin	-	unclear	Advanced solid tumors	[50,51]
	BAY1895344	Phase-I NCT03188965 clinicaltrials.gov	25 March 2022	monotherapy	-	unclear	**Solid cancers** and **lymphomas**	[52]
	AZD6738	Phase-II NCT03682289 clinicaltrials.gov	19 March 2023	+olaparib	Good oral bioavailability	yes (good)	**Renal** and **pancreatic** carcinoma, **glioma** initiating cells	[49,53]
**DNAPK**	M3814	Phase-I NCT02316197 NCT02516813 clinicaltrials.gov	19 December 2020	monotherapy +IR +cisplatin +doxorubicin	Orally bioavailable	unclear	**CLL** and **solid tumors**	[69,70]
	CC-115	Phase-I NCT02977780 clinicaltrials.gov	May 2022	+neratinib +temozolomide	-	yes (good)	**Glioblastoma multiforme**	[73]
**Chk2**	PV1019	Pre-clinical	-	+IR +topotecan	Good bioavailability	yes (good)	Human **breast cancer** (MCF7), Human **ovarian cancer** (OVCAR-3,-4,-5,-8), Human **glioblastoma** (U251)	[43]
	CCT241533	Pre-clinical	-	+bleomycin +olaparib +IR	Good bioavailability	yes (good)	Human **colorectal cancer** (HT-29), human **breast cancer** (MCF7), human **glioblastoma** (U87MG), human **ovarian cancer** line (OVCAR-3,5)	[44]
**Chk2/Chk1**	AZD7762	Phase-I NCT00473616 clinicaltrials.gov	February 2011, terminated due to cardiotoxicity	+irinotecan	-	unclear	Solid advanced tumors, glioblastoma primary isolates (pre-clinical)	[45]
**Chk1**	LY2606368	Phase-I NCT04023669 clinicaltrials.gov	June 2026	+gemcitabine +cyclophosphamide	Good oral bioavailability	unclear	Advanced **solid tumors**, **medulloblastoma**	[60]
	SRA737	Phase-I/II NCT02797964 NCT02797977 clinicaltrials.gov	28 October 2019	monotherapy +cisplatin +gemcitabine	Good oral bioavailability	unclear	Advanced **solid tumors**, **Non-Hodgkin’s lymphoma**	[59]
**Wee1**	MK-1775	Phase-I NCT02207010 NCT01849146 clinicaltrials.gov	May 2018, 28 September 2020	monotherapy +IR +temozolomide	Good oral bioavailability	yes (good)	Recurrent glioblastoma, **Glioblastoma** xenografts, Human glioblastoma cell line (U251, U87MG, T98G)	[66,67,68]
**PARP**	Rucaparib	Phase II NCT01891344 Phase III NCT01968213 clinicaltrials.gov	31 October 2021 June 2020	monotherapy	Good oral availability	yes (poor)	**Ovarian** cancer, **Epithelial ovarian** cancer, **Fallopian tube** cancer	[83,84,98,99]
	Niraparib	Phase I NCT01294735 Phase II NCT03307785	May 2012 February 2020	monotherapy +temozolomide +bevaciumab +carboplatin	Good bioavailability	yes (good)	**Melanoma**, **Glioblastoma Multiforme**, **Metastatic solid tumors**	[85]
	Veliparib	Phase I NCT01514201 Phase II NCT03581292 Phase III NCT02152982	28 March 2018, 29 October 2024, 14 January 2021	+IR +temozolomide	Good oral bioavailability	yes (good)	Newly diagnosed **Glioblastoma** and **glioma**	[86]
	Olaparib	Phase II NCT03233204 Phase II NCT02974621 Phase IINCT 03212274 Phase I NCT03212742	30 September 2024, 31 May 2020, 31 July 2020, 30 June 2022	monotherapy +bevacizumab +IR +temozolomide	Good oral bioavailability	yes (poor)	**Non-hodgkin lymphoma**, Advanced **solid tumours**, **Glioma**	[89,90]

## Figures and Tables

**Figure 1 ijms-21-04910-f001:**
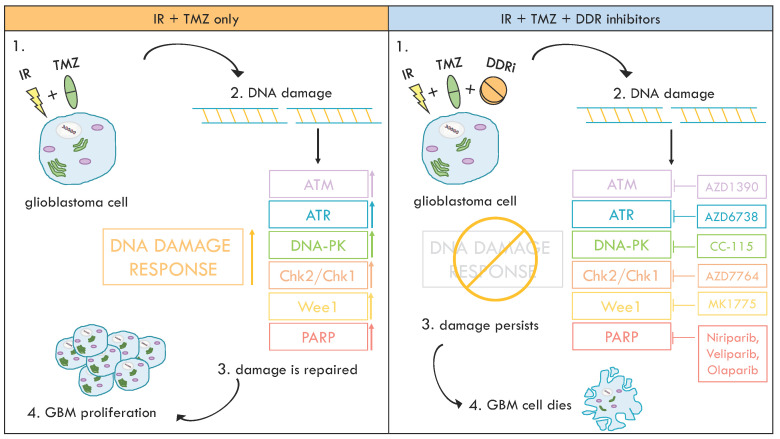
Treatment of cancerous cells with the standard combination of ionizing radiation (IR) and temozolomide (TMZ) causes DNA damage and subsequent activation of the DNA damage response (DDR) kinases (i.e., ATM, ATR and DNA-PK). Over-activation of such proteins is frequent in cancers and is responsible for therapy resistance. Addition of a DDR inhibitor to standard therapy helps reduce DNA repair rate and increases the mortality of tumor cells. Inhibitors shown in the graphic are currently been tested for glioblastoma multiforme (GBM) treatment. IR, ionizing radiation; TMZ, temozolomide.

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
