# Peer review of "Targeting the DNA Damage Response to Overcome Cancer Drug Resistance in Glioblastoma"

_ijms, 2020, doi:10.3390/ijms21144910_

Round 1

Reviewer 1 Report

Dear authors, 

I've carefully read your review manuscript entitled "Overcoming the challenges of cancer drug resistance in Glioblastoma through DNA damage kinases targeting" and I very positive on the relevance of your work and I support its publication. Given the great need to identify effective therapies for glioblastoma patients, and considered the  the limitation of the present TMZ+IR protocols, you contribution shade lights on the complicated filed of DNA damage response and its role as a drug target for GBM.

The way the manuscript is organized is very linear and logic, driving the reader to understanding the present status of the different compounds that are available at the different stages of development. There is an accurate and rich list of references that offers a broad and updated view of the present literature of the field, specially for the many compounds summarized in Table 1. I have only few comments and suggestions  to further improve your manuscript that are described below. 

1) The Title would be more sound and more appropriate if you would explicitly refer to DDR inhibitors not limiting only to kinases as you also talk about PARP.

2)There is no reference in the text to the figure 1 that should be described in the body text and not only in the figure caption.

3) Figure 1 is rather too simplified and shows only the general concept. It would be of great help if you could integrate the paper with a figure 2 (if space allows) that described a schematic view of the effect of inhibition of each protein target you consider, including PARP.

4) in the "Introduction" part you should explain better the rationale of targeting DDR and give a general description of the role of ATM, ATR, DNA-PK and PARP. This would give an easier understanding to the reader that will be more prompt to read the subsequent sections. 

5) In section 3, explain the concept of radioresistance that is an hallmark of GBM and makes this tumor so difficult to treat. 

6) In subtitle 3.1 and 3.2 add the abbreviation for SSB and DSB. 

7) I would suggest to change the title of section 4 in "TMZ treatment and cell death".

8) Reference list: I noted that at line 224 and 225 there is a reference to CHarrier et al,2011 that is not present in the final list; and that AZ20 refer to paper nr. [37], while should be [53]. Please check carefully all the numbers and relative correspondent paper. 

9) the english might be improved by the revision of a native english speaker. Some sentences could be written in a more clear and precise way, specially in the introduction part.

Author Response

Answers to the Reviewers

Reviewer 1

Dear authors,

I've carefully read your review manuscript entitled "Overcoming the challenges of cancer drug resistance in Glioblastoma through DNA damage kinases targeting" and I very positive on the relevance of your work and I support its publication. Given the great need to identify effective therapies for glioblastoma patients, and considered the  the limitation of the present TMZ+IR protocols, you contribution shade lights on the complicated filed of DNA damage response and its role as a drug target for GBM.

The way the manuscript is organized is very linear and logic, driving the reader to understanding the present status of the different compounds that are available at the different stages of development. There is an accurate and rich list of references that offers a broad and updated view of the present literature of the field, specially for the many compounds summarized in Table 1. I have only few comments and suggestions  to further improve your manuscript that are described below.

We thank the Reviewer for his/her appreciation for our work, this is very important for us as our main objective is to provide a useful tool for people working in the field. Below, we addressed point by point each comment and suggestion.

  • The Title would be more sound and more appropriate if you would explicitly refer to DDR inhibitors not limiting only to kinases as you also talk about PARP.

Following this suggestion, the Title has been revised. The previous title “Overcoming the challenges of cancer drug resistance in Glioblastoma through DNA damage kinases targeting” has been replaced with a new one “Targeting the DNA damage response to overcome cancer drug resistance in Glioblastoma”

  • There is no reference in the text to the figure 1 that should be described in the body text and not only in the figure caption.

We apologize for the mistake. The reference to Figure 1 has also been included in the text as suggested by the Reviewer.

  • Figure 1 is rather too simplified and shows only the general concept. It would be of great help if you could integrate the paper with a figure 2 (if space allows) that described a schematic view of the effect of inhibition of each protein target you consider, including PARP.

We reasoned carefully on how to accomplish this request by the reviewer. We decided to modify Figure 1 in order to provide a more complete view of the effect of the inhibition of each protein target we considered in this review. In order to be focused on the main objective of this review we point the attention on those compounds that have been included in clinical trials in glioblastoma. We hope that this revised version of the figure may be more informative for the readers.

  • in the "Introduction" part you should explain better the rationale of targeting DDR and give a general description of the role of ATM, ATR, DNA-PK and PARP. This would give an easier understanding to the reader that will be more prompt to read the subsequent sections.

We added a short paragraph in the Introduction to provide already in this section a first description of the role of ATM, ATR, DNA-PK and PARP that may be helpful for the readers to better understand the following sections.

  • In section 3, explain the concept of radioresistance that is an hallmark of GBM and makes this tumor so difficult to treat.

According to this reviewer, we added a paragraph on radioresistance and chemoresistance in GBM. This paragraph has been inserted at the end of section 2.

6) In subtitle 3.1 and 3.2 add the abbreviation for SSB and DSB.

We modified the subtitles according to this suggestion.

7) I would suggest to change the title of section 4 in "TMZ treatment and cell death".

We modified the tile of section 4 according to this suggestion.

8) Reference list: I noted that at line 224 and 225 there is a reference to CHarrier et al,2011 that is not present in the final list; and that AZ20 refer to paper nr. [37], while should be [53]. Please check carefully all the numbers and relative correspondent paper.

We are grateful to the Reviewer for his/her attention and we carefully checked the references and highlighted the corrections.

9) the english might be improved by the revision of a native english speaker. Some sentences could be written in a more clear and precise way, specially in the introduction part.

We revised the text and in particular the Introduction, according to this suggestion.

Reviewer 2 Report

This excellent review summarizes the possibilities of DNA damage response (DDR) kinase inhibitors in glioblastoma. The molecular background of DDR mechanisms is nicely explained. Then, the authors present the existing data for DDR kinase inhibitors in glioblastoma. All relevant work is cited. The linguistic style is very good, making the manuscript easy to read and understand. This work will be of high interest to your readers. Therefore I strongly recommend publication without further changes needed.

Author Response

Answers to the Reviewers

Reviewer 2

This excellent review summarizes the possibilities of DNA damage response (DDR) kinase inhibitors in glioblastoma. The molecular background of DDR mechanisms is nicely explained. Then, the authors present the existing data for DDR kinase inhibitors in glioblastoma. All relevant work is cited. The linguistic style is very good, making the manuscript easy to read and understand. This work will be of high interest to your readers. Therefore I strongly recommend publication without further changes needed.

We are pleased that the Reviewer appreciate our work and we are happy to hear that he/she believes that our manuscript will be of high interest to readers in the field of glioblastoma and DNA damage response.
